# Current Challenges in Coronary Bifurcation Interventions

**DOI:** 10.3390/medicina60091439

**Published:** 2024-09-03

**Authors:** Panayot Panayotov, Niya Mileva, Dobrin Vassilev

**Affiliations:** 1Department of Cardiology, Pulmonology and Endocrinology, Medical Faculty, Medical University of Pleven, 5800 Pleven, Bulgaria; 2Medica Cor Hospital, 7013 Ruse, Bulgaria; nmileva91@gmail.com (N.M.); dobrinv@gmail.com (D.V.); 3Faculty of Public Health and Healthcare, Ruse University “Angel Kanchev”, 7017 Ruse, Bulgaria

**Keywords:** percutaneous coronary intervention, coronary bifurcation, optimization technique

## Abstract

Coronary bifurcation lesions account for a significant proportion of all percutaneous coronary interventions (PCIs). Interventional treatment of coronary bifurcations is related to significant technical challenges, high complication rates, and worse angiographic and long-term clinical outcomes. This review covers the specific features and structure of coronary bifurcation and explores the main challenges in the interventional treatment of these lesions. This review evaluates various methodologies designed to address these lesions, considering factors such as plaque distribution and bifurcation geometry. It also emphasizes the limitations associated with current techniques. A novel combined optimization approach applied in the interventional treatment of coronary bifurcation may offer superior procedural and long-term outcomes. This combined technique could potentially address the drawbacks of each method, providing a more effective solution for optimizing stent placement in bifurcation lesions. Refining and evaluating these combined techniques is essential for improving clinical outcomes in patients with bifurcation lesions.

## 1. Introduction

Coronary bifurcation lesions account for nearly 20–25% of all percutaneous coronary interventions (PCIs) [1,2]. Procedures for this subset of lesions present a technical challenge with a high incidence of early and late complications [3]. PCIs in bifurcation stenoses can be performed using a variety of techniques, depending on the distribution of plaque along the main and stent branches and the geometry of the bifurcation [4]. The geometry of coronary bifurcations determines the dimensional discrepancy in diameters between the proximal main vessel (MV) and the daughter branches and the distal main vessel (main branch, MB) and the side branch (SB) [5]. Several optimization techniques are available to improve procedural outcomes in the treatment of these compressive lesions. The kissing balloon inflation (KBI) technique was one of the first stent optimization techniques proposed specifically for bifurcation lesions, and it continues to play an important role in bifurcation PCIs by optimizing stent placement and improving side branch access. However, the use of KBI requires crossing the side branches after stenting the main vessel, which adds additional procedural and fluoroscopic time as well as contrast volume. Some degree of operator experience is also required, especially in cases of SB occlusion after stenting. Additional disadvantages of KBI include elliptical deformation of the proximal main vessel, which may further compromise long-term outcomes [6]. The proximal optimization technique (POT) has been proposed as a stent optimization technique that can adapt the tubular design of the coronary stent to the natural anatomy of the bifurcation [7]. It was expected that the POT could adjust the stent deployment by conforming to the fractal anatomy of the vessel without compromising SB patency or even improving it. However, studies have shown that additional balloon dilatation of the SB is necessary to maintain SB patency without compromising functional vascular flow [8]. The optimal outcome of the POT is highly dependent on accurate balloon positioning, and inaccurate balloon placement may result in incomplete string adhesion to the lateral wall of the SB [9,10]. In addition, data have shown that even a correctly positioned POT balloon (according to current criteria [4]) can cause additional elliptical deformation in the SB ostium, resulting in further stenosis [11]. Optimization techniques after stent placement in a patient with a bifurcation lesion need to be refined. A technique consisting of a combination of KBI and the POT may result in better procedural outcomes and better long-term clinical outcomes in these patients. The current review aims to discuss the specific aspects of the coronary bifurcation lesions and explores the main challenges in the complex process of the bifurcation PCI [12].

## 2. The Coronary Bifurcation Structure

The arteries providing the blood supply to the myocardium were named “coronary arteries” due to their visual resemblance to a “corona” that covers the cardiac myocardium. The coronary arteries are structured like a tree with dichotomous branches. In practice, there are an innumerable number of branches starting from the coronary ostia. This structure allows for an even distribution of nutrients and oxygen to each unit of myocardial tissue. The structure of the coronary circulation follows Murray’s Law, which states that the radii of the daughter branches are related to the radius of the parent branch (Figure 1). Therefore, at any branching point in the coronary tree, the diameter of the vessel can be predicted. Professor Gérard Finet simplified Murray’s law and showed that the diameter of the proximal main vessel is approximately equal to the sum of the diameters of the daughter vessels (distal main plus lateral) multiplied by 0.67.

Therefore, in each bifurcation, the diameter of the distal main vessel will be consistently smaller than the diameter of the proximal main vessel. The larger the diameter of the lateral branch, the greater the difference between the diameter of the proximal main vessel and that of the vessel distal to the bifurcation. This difference in diameters is important in selecting the appropriate stent size. If a stent is selected based on the diameter of the proximal main vessel, it will be “too large” for the distal main vessel.

On the other hand, blood flow in the coronary arteries is determined by the general laws of hydrodynamics. It is accepted that the general structure of the branches of the arteries, called bifurcations, follows the principles of optimality in physics. That is, the geometry of coronary bifurcations is designed to minimize the forces of resistance as well as the total surface area or volume of the bifurcation [1,2,3,4,5]. Later, it was found that there are deviations (sometimes very significant) from optimality and that certain geometric factors may be responsible for the accumulation of atherosclerotic plaques at bifurcation sites [6,7,8]. As originally proposed by Caro and Fitz-Gerald [9], low frictional stress (sheer stress) conditions exist in bifurcation regions. This leads to areas of slower blood flow, and as a result, a boundary layer of stagnant blood is formed. This boundary layer typically forms near the outer walls of the branches, opposite to the flow divider (also called the carina) [10,11]. Compensatory expansion (positive remodeling) or contraction of the vessel further increases the complexity of the bifurcation structure [13,14]. On the other hand, high frictional stresses, increased endothelial cell renewal, and thus a low probability of plaque accumulation are observed at the site of the carina [15,16,17]. All of these factors favor the formation of bifurcation stenoses with a very complex structure.

## 3. Epidemiology

Coronary bifurcation lesions are a relatively common finding in interventional practice, accounting for between 12 and 20% of all cases treated [18,19,20,21,22,23]. The NHLBI registry included 2436 patients and obtained information on interventional practice in the late 1990s [19]. Of these, 321 patients (13%) were treated for bifurcation stenosis. Although this study does not reflect state-of-the-art interventional practice (approximately 60% use of bare metal stents (BMS)), it provides insight into the importance of the bifurcation lesion as a pathology. Complete angiographic success was achieved in only 86% (93.5% in non-bifurcation lesions) of patients with a higher incidence of side branch occlusion than in non-bifurcation cases (7.3% vs. 2.3%), and, most importantly, a higher incidence of major adverse cardiovascular events (MACEs)—death, myocardial infarction, and surgical revascularization—in the first year was observed (7.2% vs. 5.0%, bifurcation–non-bifurcation group). The PRESTO trial [20], a large study investigating the effect of tranilast on restenosis rates, indicated that of 10,068 patients enrolled, 1412 had bifurcation stenosis (14%). Patients with bifurcation lesions had higher rates of significant residual stenosis after the procedure (7.7% vs. 5.1%, bifurcation group–non-bifurcation group) and higher rates of dissections (16% vs. 13%, non-bifurcation group). Patients with bifurcation lesions had a higher rate of target lesion revascularization (TVR), 21% vs. 17%, bifurcation vs. non-bifurcation, respectively, and a higher rate of target lesion restenosis (TLR), 28% vs. 22%, bifurcation versus non-bifurcation groups, respectively. Interestingly, the overall risk profile in the bifurcation group was better than in the non-bifurcation group (34% vs. 39% fewer previous myocardial infarctions; 11% vs. 14% fewer previous CABG; 4% vs. 6% less peripheral artery disease), but there was a higher incidence of stable angina (44% vs. 41%, bifurcation versus non-bifurcation groups). These facts and the generally lower risk profile of the population in this study (mean ejection fraction of 61%) may explain the lack of difference in mortality and MI at one year (less than 1% in both groups). However, bifurcation per se is associated with a higher risk of percutaneous coronary intervention (PCI), a higher incidence of intraprocedural complications, and a higher incidence of TVR and TLR at follow-up.

In more recent interventional studies with the introduction of drug-eluting stents (DES), patients with a bifurcation lesion retain their higher risk profile. In the SCANDSTENT trial [21], 322 patients with complex lesions and a high-risk profile (chronic total occlusions, bifurcations, ostial lesions, and angulated lesions) were randomized: 163 patients received a drug-eluting stent (DES), and 159 patients received a bare metal stent (BMS). Bifurcation lesions occupied 35% and 33% of both groups, respectively. The incidence of TLR in the DES group was related to bifurcation lesions (7.1% in the DES group vs. 29.4% in the BMS group, *p* < 0.001, non-bifurcation lesions vs. bifurcation lesions). Interestingly, drug-eluting stents significantly improved patients’ clinical conditions due to a significant reduction in late lumen loss (0.12 mm main branch in the DES group vs. 99 mm in the BMS group; side branch 0.03 mm DES vs. 0.56 mm BMS) and restenosis rate (main branch 4.9% DES vs. 28.3% BMS; side branch 14.8% DES vs. 43.4% BMS), and as a result, the incidence of MACE was significantly lower, at 9% vs. 28% at follow-up [22]. In the ARTS II trial, 607 patients with multivessel disease were treated with sirolimus-eluting stents [23]. Of these, 324 patients had bifurcation stenosis (53%). Again, the group with bifurcation had a higher risk profile with a significantly greater number of vessels affected, a larger number of lesions (3.9 vs. 3.2), and a higher incidence of long and diffuse lesions (14.2% vs. 8.7%). Treatment of bifurcation lesions was associated with a longer procedural time (92.6 min vs. 76.5 min), a higher number and length of stents, and more frequent application of gpIIB/IIIA (36.7% vs. 27.6%). The authors state that the clinical course of these patients was the same as those without a bifurcation lesion. However, a closer look at the study results shows that there was a very strong trend toward an increase in the number of non-fatal myocardial infarctions (5.2% vs. 2.5%, bifurcation vs. non-bifurcation group, *p* = 0.1) at the 30-day follow-up. The same trend persisted at the 1-year follow-up, with a higher incidence of myocardial infarction (5.9% vs. 2.8%, *p* = 0.08) and death/CVA/MI (7.1% vs. 4.2%, *p* = 0.16), as well as a higher rate of repeat revascularization (7.7% vs. 4.9%). It is clear from the above discussion that bifurcation lesions represent a significant clinical problem. They are clearly a marker of more advanced coronary disease and are associated with lower interventional success, higher periprocedural complications, and worse long-term clinical outcomes.

There are two main problems in the treatment of bifurcation lesions. The first is the compromise of the side branches, manifested by vessel closure or significant ostial stenosis after main vessel stenting, compromising blood flow in the side branch. The second is the higher incidence of restenosis, which leads to a higher incidence of adverse events in the long term, recurrent symptoms, and acute and late stent thrombosis.

## 4. Side Branch Compromise

There is still uncertainty about the mechanisms of SB compromise after main vessel stenting. Currently, there are two main hypotheses: (1) plaque displacement from the main to the side branch and (2) carina displacement due to carina tip extrusion into the side vessel ostium [1]. Based on phantom elastic patterns and then an angiographic analysis of a cohort of patients, it was concluded that carina displacement is probably the most important mechanism of side branch compromise [3]. However, in addition to possible plaque displacement, there is another potential mechanism, namely ostial deformation, which leads to a change in the shape of the lateral vessel ostium from an initially ostial circle to an ostial ellipse after stenting [11]. More than a decade ago, deformation of the pre-stenting circular ostium into a post-stenting elliptical orifice was proposed as a mechanism for SB compromise based on theoretical assumptions and observations from phantom elastic models of coronary bifurcations [2,3]. It can be hypothesized that the elliptical shape of the ostium is the last common pathway that occurs in the ostium after the stenting of the main vessel. Elliptical stretching and deformation could occur and explain ostial side branch stenoses (even of high degree) in branches arising at 90° from the main vessel where carina displacement is theoretically impossible. Consistent with the assumptions of an initial circular minimum diameter of the lateral branch lumen that transforms into an ellipse are data from the literature demonstrating that almost 90% of bifurcations have circular ostia [5,6]. In recent years, these theoretical and experimental observations have been confirmed by optical coherence tomography of coronary bifurcations after MI stenting [5,6].

A study conducted by Vassilev et al. provides a quantitative basis for calculating the area stenosis based on angiographic data. The formulas used were adopted from the literature [8,9,10,11,12,13,14,15] and were limited based on the assumption of a constant vessel circumference. Ramanujan’s formula was used [8,9,13], which provides values closest to the mean of all other formulas. The area of stenosis calculated based on elliptical assumptions provides a much more physiological area of ostial stenosis in better agreement with experimental observations of flow limitation caused by stenosis [16,17]. This observation is more significant for less severe stenoses (<70% area stenosis) where the differences in area are greater, whereas for more severe stenoses (>70% area stenosis), the shape of the ostium does not seem to be as important, and the area stenosis values are more circular, regardless of which formula is used. In addition, the study demonstrated that ostial area stenosis calculated using this assumption correlated better with the physiological parameter of lesion severity—i.e., FFR—than area stenosis calculated using the traditional circular formula. It should be noted that the calculations in this study were based on the assumption of the ability of the SB ostium to deform freely, as quantified by the stretch factor k. In fact, the presence of plaque with fibrous content and calcium can prevent these deformations [18].

### 4.1. The Balloon Angioplasty Era

Since the first attempts to intervene with bifurcation lesions, several reports on the incidence of side branch occlusions in balloon angioplasty have been published [24,25,26]. In the study by Meier et al. [24], which was the first to address this issue with a large group of patients (*n* = 302 patients; 54% of all patients underwent PTCA during the study period), the incidence of side branch occlusion was 5% overall for the study group, 14% for branches originating from a segment of stenosis, and only 1% in the group without ostial narrowing. The course of the disease in the patients was relatively benign, with angina as the main complication. Vetrovec et al. [25] studied 76 patients with 97 side branches and found a 9% incidence of occlusion. In particular, in patients with ostial branch stenosis before PTCA, the occlusion rate was 27%, whereas in the group without ostial involvement, the occlusion rate was only 4%. In the study by Boxt et al. [26], 17% of 93 side branches (86 patients) were occluded after PTCA. This is the first study of a larger group of patients to show reopening of occluded lateral branches in more than 70% of them.

In the study by Weinstein et al. [27], 56 patients (2.5% of all patients treated at the authors’ center during the study period) underwent balloon angioplasty of both vessels (63%) or only the main vessel (37%). The overall incidence of temporary SB occlusion after main vessel dilatation was 34% (32% in both groups). In the first group, 91% of the occluded branches were successfully opened, whereas in the second group, only 38% of the occluded branches could be opened, and 24% remained completely occluded. The authors explained the higher occlusion rate in their series by higher ostial stenosis in the SB (all branches had >70% ostial stenosis), all with a diameter greater than 1.5 mm. Notably, 37% of patients in the second group with a higher number of permanently occluded branches remained symptomatic and had signs of ischemia at discharge. It is evident from the above results that in the PTCA era, the average incidence of SB occlusion is 5–10%. However, these obstructions mostly affect small-caliber vessels, most commonly around 1 mm. The treatment rate of bifurcation lesions involving large SBs is very low, 2.5–3% of all lesions treated [28,29,30]. If the branches were less than 1.5 mm in diameter and had ostial involvement, the occlusion rate increased to 30–40% but with a lower incidence of myocardial infarction, prolonged ischemia, and chest pain.

### 4.2. The Stent Era

With the introduction of stents in the early 1990s, the spectrum of lesions treated increased greatly [31,32]. Results from the STRESS and BENESTENT trials showed significantly lower restenosis rates mainly by achieving greater lumen enlargement. With the introduction of drug-eluting stents (DES), a new era in interventional cardiology had begun [33,34,35,36,37], with dramatic reductions in restenosis rates in all lesion types. However, there has been a trend of a greater periprocedural increase in cardiac enzymes and an alarming trend toward higher mortality [38,39,40,41,42]. The Cleveland Clinic investigators [39] studied a group of 8409 patients undergoing elective PCI and analyzed mortality trends after PCI at 4 months and 1 year. They found a significant difference between the levels of CK-MB elevation after stent implantation and after POBA (in the death group, 71% of patients were stented vs. only 25% with POBA).

The same trends were observed in two other larger studies. In the study by Stone et al. [43], 7147 elective patients were followed for 2 years. The incidence of periprocedural non-Q myocardial infarction (defined as more than a 3-fold increase in CK-MB) was 17.9%. Increased mortality was demonstrated when CK-MB was elevated more than eight times normal or when Q waves were detected in the surface ECG. The incidence of non-Q MI was 16.5% vs. 12.1% for the stent and balloon procedures. The 2-year absolute survival difference was more than 6% between patients with enzyme elevations below and above 8 times the normal increase. Herrmann et al. [44] systematically examined differences in enzyme release after PCI with different devices and the reasons for this in a group of 1675 consecutive patients. Several factors were associated with an increase in CK-MB-sudden vessel occlusion, distal thromboembolism, slow flow, prolonged hypotension, persistent chest discomfort after the procedure, and, very commonly, side branch closure. Balloon angioplasty was associated with an 11.5% vs. 21% increase in CK-MB. Side branch occlusion was the most common procedural complication (23% in the group with elevated enzymes). It was more common with stent use (36% vs. 30% with POBA in the elevated enzyme group). The incidence of side branch closure was similar in the absence of elevated cardiac biomarkers (13% vs. 11%, POBA and stent, respectively). Investigators from the REPLACE-2 trial [45,46] analyzed 10 types of adverse angiographic events in a cohort of 6010 patients. The rate of increase in CK-MB was 24.3% in the overall group but 47% in the adverse event group. The strongest predictor of death, MI, and TVR was an adverse angiographic event composed of four different events (OR = 1.9, CI 1.6–2.4, *p* < 0.001). The events of side branch closure, sudden occlusion, slow blood flow, and distal embolization predicted increased mortality at 6-month follow-up. A very important finding was that transient side branch occlusion was an independent predictor of increased CK-MB in multivariable with a likelihood ratio of 4.54 (CI 1.02–20.21, *p* = 0.047) and occurred in 4.4% of all cases [47]. In all of the above studies, creatine kinase was used to detect myocardial necrosis. However, it is not as sensitive as the troponin test, and it has been shown that in the group of patients with PCI in whom CK-MB was elevated in 16% of cases, troponin T was elevated in 60% at the same time [48]. This implies that the above values may underestimate the true significance of periprocedural enzyme elevation, which results from collateral branch compromise (as an aggregate of closure and flow disruption). In the study by Natarajan et al. [49], 17% of 1128 patients had elevated troponin I with no change in CK-MB. More than half (53%) of them had a value more than five times the reference value. Side branch closure was a major predictor of such elevation (OR 3.0, CI 1.4–6.6, *p* < 0.005) and was a predictor of 1 year MACE(d) (death, nonfatal myocardial infarction, and repeat revascularization)—OR 1.6, CI 1.0–2.5, *p* = 0.05.

It must be concluded that the side branches require special attention, as their compromise is associated with myocardial necrosis, which worsens the patient’s prognosis [50]. In view of the above, several studies have attempted to identify predictors of side-branch compromise. Several factors have repeatedly emerged as predictors of ostial stenosis and occlusion. They are as follows:-Occurrence of main vessel branch stenosis [51,52,53,54,55,56,57,58,59,60,61,62,63,64];-Stenosis in the ostium of the branching vessel prior to implantation of a parent vessel stent [61,62,63,65,66,67,68,69,70,71,72,73,74];-Small reference diameter side branch [69,75,76,77,78];-Higher implantation pressure [64,66,79,80,81];-Higher balloon-to-arterial ratio in the main vessel [71,72,73,80];-Balloon predilatation compared with direct stent implantation [72,73,76,81,82,83,84,85];-Distal angles between branches [66,72,86,87].

The main problem that remains after all these studies is the lack of predictability of side-branch compromise. A better understanding of the factors that lead to side-branch compromise will help to find a better therapeutic solution when such complications occur. For example, most of the factors that determine side-branch stenosis after main-branch stenting become predictors of late restenosis because, as known from the early years of PTCA, a smaller minimum lumen diameter leads to higher restenosis rates [84]. A study by Koo et al. [88] using fractional flow reserve (FFR) measurements showed that ostial stenosis in the SB after main vessel stenting must be greater than 85% in diameter to restrict blood flow. This finding is consistent with previous observations of relative benignity when the ostial branch stenosis is small in diameter. However, it does not explain why such a large stenosis does not result in impaired blood flow.

## 5. Restenosis in Bifurcation Lesions

There are no large prospective randomized trials evaluating long-term clinical outcomes in a population with bifurcation lesions after BMS placement vs. balloon dilatation or surgery or with different stenting techniques. Most information is based on registry reports and retrospective data. Outcome data after stenting with BMS are summarized in Table 1. In the studies described, the technical success rate for main and side branch dilatation was greater than 87% in most cases, but the clinical outcomes obtained remained heterogeneous. The restenosis rate was 25–62% in the two-stent group vs. 12.5–48% in the single-stent group, and the target lesion revascularization (TLR) rate was 24–43% in the two-stent group vs. 8–36% in the single-stent group. The study by Leferve et al. deserves special attention because it represents the largest reported series of patients in which the causes of target lesion revascularization have been studied in detail [89]. The authors identified several independent predictors of major adverse cardiovascular events (MACE)—unstable angina, proximal vessel diameter <2.7 mm, use of 2 stents, and type 1 and type 4 lesions (bifurcations are classified according to ICPS Paris Sud classification as type 1 if there is significant stenosis before, after, and at the SB ostium, type 2 if there is significant stenosis before and after the SB, type 3 if there is significant stenosis before the SB, and type 4 if the MB after the SB and the SB ostium are significantly stenosed, with subtypes a if the MB alone is affected and b if the SB ostium alone is affected). In type 4 lesions, the incidence of MACE events at follow-up was 29%, whereas in type 3 lesions, it was less than 9%. It should be noted that type 1 and type 4 lesions denote the origin of a side branch from a lesion of the main vessel and ostial branch vessel stenosis. These are the strongest predictors of side branch compromise after main branch stenting, which also predicts periprocedural complications. In the TULIPE study [76], again, one of the three independent predictors of a subsequent MACE event was side branch stenosis, branch calcification, and left ventricular function.

The results for DES are shown in Table 2. The overall success rate was 94–100%. Restenosis rates ranged from 5.1 to 28% in the two-stent group vs. 5.3 to 18.7% in the single-stent group, and TLR ranged from 1.0 to 31.1% in the two-stent group vs. 1.9 to 5.4% in the single-stent group. This was also associated with late stent thrombosis, ranging from 0.5 to 4.3%. The Nordic Bifurcation Study [87] compared the strategy of stenting the main vessel and side branch with stenting of the main vessel only, with optional stenting of the side branch with sirolimus-eluting stents. At six months, there were no significant differences in MACE between the two groups. However, stenting of both the main vessel and the side branch was associated with significantly longer procedure and fluoroscopy times, greater contrast volumes, and a higher rate of procedure-related increases in biomarkers of myocardial injury (8% versus 18% in the one-stent vs. two-stent group, respectively). However, this did not translate into worse outcomes in this group, with one of the largest discrepancies found between restenosis rates and revascularization rates (less than 2%). 

Overall, the data on the use of DES showed a trend toward less restenosis at long-term follow-up compared to BMS. Stenting both the main vessel and side branch provides better visual satisfaction but increases the propensity for subacute stent thrombosis in the early follow-up period. The single stent technique is superior to main vessel and side branch stenting in terms of procedural success, fluoroscopy time, and contrast volume, with an overall reduction in TLR. Again, predictors of side-branch compromise leading to a worse post-procedural outcome are a higher incidence of restenosis and revascularization [79]. In the SIRIUS bifurcation study [59], 53% of patients switched from a single-stent technique to a two-stent technique due to an unsatisfactory primary outcome. Lateral branch stenosis was higher in the two-stent group (56% vs. 46%), leading to a long-term increase in restenosis (22% vs. 14%—two-stent vs stent/balloon groups) with almost identical rates of main vessel restenosis, 5.7% and 4.8%, for the one-stent group. It is reasonable to assume that a worse procedural outcome associated with a higher side branch compromise leads to worse long-term outcomes. Iakovou et al. [84] demonstrated a continuous risk of stent thrombosis on a continuous scale, with a 5.96-fold higher risk of subacute thrombosis and an 8.11-fold higher risk of late stent thrombosis. It is important to note that the risk does not decrease over time but actually increases. Apparently, complex hydrodynamic conditions are predisposed to local stasis and thrombosis, and this is more evident with more complex approaches [84,89,90].

It is evident from the above data that side branch compromise is a significant clinical problem in percutaneous treatment of bifurcation lesions. It complicates the hospital and long-term condition of patients. There is uncertainty about the exact pathophysiological mechanism and predictors of its occurrence.

## 6. Bifurcation Optimization Techniques

The desire to achieve an optimal outcome in the interventional treatment of bifurcation lesions has stimulated the development of various optimization techniques to promote a better outcome with a lower incidence of side branch compression and a lower incidence of in-stent restenosis.

### 6.1. The Kissing Balloon Inflation (KBI) Technique

The final KBI procedure consists of positioning balloons in the main vessel (MV) and the side branch (SB), after which they are inflated simultaneously. This maneuver is thought to be effective after MV cross-stenting to ensure SB patency, reduce SB stenosis, and remove stent struts that are jammed (jailed struts) (Figure 2).

KBI application, however, requires side-branch crossing after the stenting of the main vessel, which adds additional procedural and fluoroscopic time as well as more contrast material. Some operator experience is also required, especially in cases with SB occlusion after stenting. An additional disadvantage of KBI is the elliptical deformation of the proximal main vessel, which may further compromise long-term outcomes [11].

Clinical outcomes from recent studies comparing KBI versus non-KBI are listed in Table 3.

Most studies have failed to show the advantage of KBI over non-KBI treatment in terms of major adverse cardiac event (MACE) rates, although SB stenosis is reduced at 6 to 12 months follow-up 2–5. In the Korean Coronary Bypass Stenting (COBIS) registry, KBI was associated with a higher incidence of MACE. On the other hand, a small observational study of acute coronary syndrome reported the effectiveness of KBI in relation to MACE (*n* = 251, 8.2% vs. 20.3%, hazard ratio 0.398, 95% CI: 0.190–0.836, *p* = 0.015). The COBIS II registry showed that SB occlusion after MV stenting was associated with more frequent cardiac death and myocardial infarction (*n* = 2227, hazard ratio 2.34, 95% CI: 1.15–4.77, *p* = 0.02), suggesting the importance of KBI in ensuring SB patency.

### 6.2. The Proximal Optimization Technique (POT)

The Proximal Optimization Technique (POT) was proposed by Dr. Olivier Daremont as a technique to compensate for this difference in diameter when stenting bifurcation lesions. The Proximal Optimization Technique (POT) was proposed as a stent optimization technique that could adapt the tubular design of the coronary stent to the natural anatomy of the bifurcation [7] (Figure 3). The technique is fully supported by the European Bifurcation Club and is a key part of the consensus statement produced by this group of experts [1]. A short balloon is expanded in the proximal master vessel just adjacent to the carina. This allows for full expansion and complete stent placement in the proximal main vessel. In addition, the POT also facilitates re-passage into the lateral branch to optimize the final outcome, if needed. It was anticipated that the POT could adjust stent application by conforming to the fractal anatomy of the vessel without compromising or even improving SB patency. However, studies have shown that additional balloon dilatation of the SB is necessary to preserve SB patency without compromising functional vascular flow [8]. The PROPOT study is a randomized clinical trial comparing the effectiveness of the POT vs. KBI, comparing stent strut apposition assessed with optical coherence tomography (OCT) [91]. A total of 120 patients were enrolled in the study, and OCT was performed at baseline, immediately after side branch re-crossing, and at the final follow-up. The incidence of malapposed struts was not significantly different between the two groups—proximal in the stent: 10.4% vs. 7.7%, *p* = 0.33; in the bifurcation: 1.4% vs. 1.1%, *p* = 0.67; distal edge: 6.2% vs. 5.3%, *p* = 0.59, POT vs. KBI, respectively). At the one-year follow-up, only one patient per group had undergone target lesion revascularization.

The optimal outcome of the POT depends largely on precise balloon positioning, and inaccurate balloon placement may result in incomplete string adherence to the lateral wall of the SB [9,10]. Moreover, data have shown that even a properly positioned POT balloon (according to current criteria [4]) can cause additional elliptical deformation in the SB ostium, thereby further stenosing it, as was demonstrated by Vassilev et al. [11].

## 7. Conclusions

Coronary bifurcation lesions account for a significant proportion of all percutaneous coronary interventions (PCIs) and are associated with significant technical challenges and high complication rates. The main issues in the treatment of bifurcation lesions are side branch compromise and restenosis rates. Several optimization techniques have been developed to improve procedural and clinical outcomes after bifurcation PCI. A novel combined optimization approach to the interventional treatment of coronary bifurcations may offer superior procedural and long-term outcomes. The refinement and evaluation of optimization techniques during bifurcation PCI may be essential to improve clinical outcomes in this subset of high-risk patients.

## Figures and Tables

**Figure 1 medicina-60-01439-f001:**
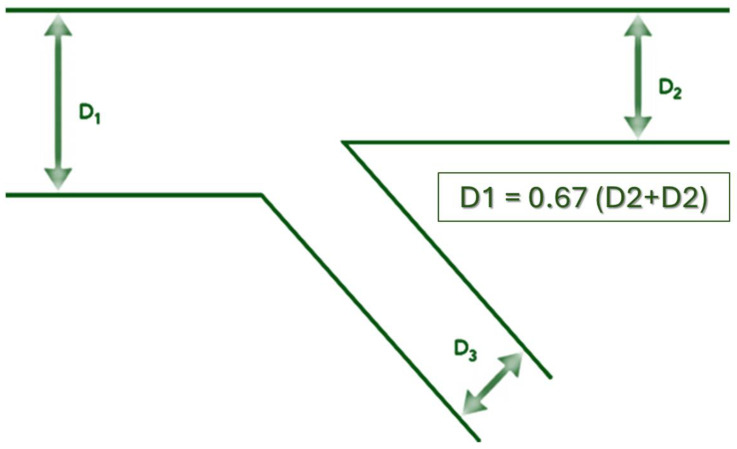
Coronary bifurcation structure: D_1_—main vessel diameter; D_2_—main branch diameter; D_3_—side branch diameter.

**Figure 2 medicina-60-01439-f002:**
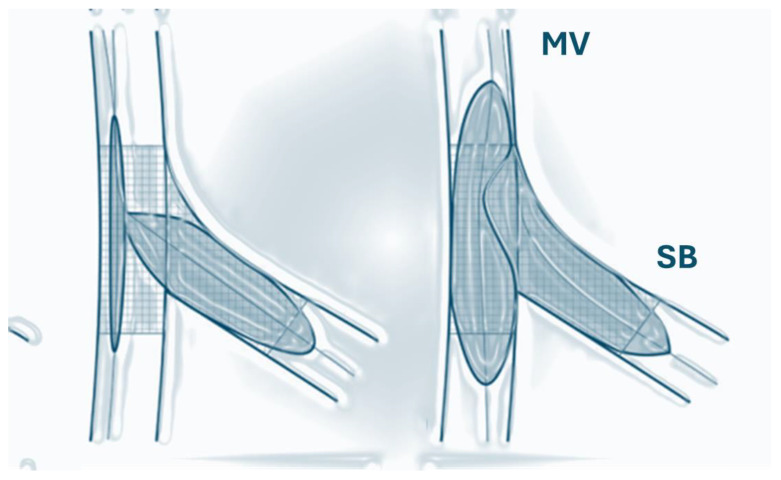
Kissing balloon inflation (KBI) technique. MV—main vessel; SB—side branch.

**Figure 3 medicina-60-01439-f003:**
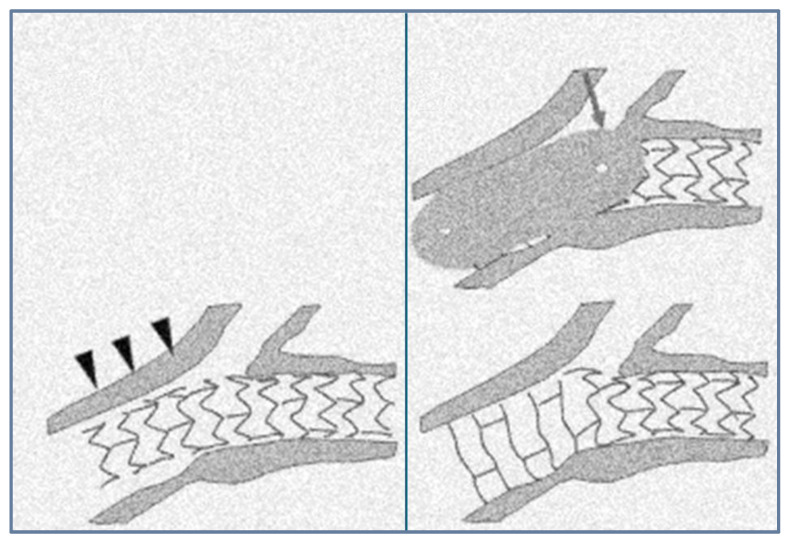
Proximal optimization technique (POT). The grey arrow annotates the ballon for optimization. Black arrows annotate stent strut apposition.

**Table 1 medicina-60-01439-t001:** Studies of coronary bifurcation lesions treated with bare metal stents. S—stent; B—balloon; PTS—provisional stenting.

Author	Randomization	Technique	Number of Patients	Follow-Up (Months)	Restenosis Rate, %	TLR (%)
Suwaidi	No	S + S	77	12	-	19.4
		S + B	54		-	20.5
Chevalier	No	S + S	50	6	28	24
Yamashita	No	S + S	53	6	62	38
		S + B	39		48	36
Pan	No	S + S	23	18	43	39
		S + B	47		19	17
Anzuini	No	S + S	45	12	25	35.5
		S + B	45		12.5	15.5
Brunel	No	S + S	50	6	57	43
		S + B	56		21	8
Brunel	No	PTS	186	7	MB—9.1	15.9
					SB—25.3	
Rux	No	S + B	370	6	13	15.7
		S + S	45		16.2	33.3
Beijk	No	S + B	465	12	11.3	8.6
		R stent				
Lefevre	No	S + B	724	7	MB/SB -	13.6
					7.3/7.3	
		S + S	425		MB/SB -	24.1
					3.2/11.5	

**Table 2 medicina-60-01439-t002:** Studies of coronary bifurcation lesions treated with drug-eluting stents. S—stent; B—balloon; PTS—provisional stenting.

Author	Randomization	Stent	Technique	Number of Patients	Follow-Up (Months)	Restenosis Rate, %	TLR (%)
Colombo	да	SES	S + S	63	6	28	9.5
			S + B	22		18.7	4.5
Pan	да	SES	S + S	47	11	20	5
			S + B	44		7	2
Ge	не	SES	S + S	117	9	24	8.9
			S + B	57		10	5.4
Ge	не	SES + PES	S+ S	181	9	MB—11.5	14.9
						SB—21.6	
Sharma	не	SES	S+ S	200	8		4
Hoye	не	SES + PES	S+ S	23	9	MB—18.8	5.3
						SB—12.5	
Hoye	не	SES + PES	S+ S	231	9	MB—9.1	9.7
						SB—25.3	
Ge	не	SES + PES	S + S − T	61	12	13	11.1
			S + S − CR	121		16.2	14
Moussa	не	SES	S + S − CR	120	6	11.3	11.3
Steigen	да	SES	S + S	206	8	MB/SB –	1.0
						5.1/11.5	
			S + B	207		MB/SB –	1.9
						4.6/19.2	
Ferenc	да	SES	S + B + S	101	12	MB/SB –	10.9
						7.3/7.3	
			S + S − T	101		MB/SB –	8.9
						3.2/11.5	
De Mario	не	PES	S + S	118	12	16.5	18.3
			S + B	32		13.1	21

**Table 3 medicina-60-01439-t003:** Studies reporting clinical outcomes comparing KBI vs. non-KBI. KBI—kissing balloon inflation technique; non-KBI—non kissing balloon inflation technique.

Study	Design	Patients, *n* KBI Non-KBI		Follow-Up (Months)	CVD KBI vs. Non-KBI	MI	TVR	Stent Thrombosis	MACE	SB Stenosis, %
Provisional stenting										
THUEBIS	Random-ized	56	54	6	0% vs. 3.7%	3.6% vs. 1.9%	18% vs. 15%	3.6% vs. 1.9%	23.2% vs. 24.1%	37% vs. 32%
Nordic III	Random-ized	238	239	6	0.8% vs. 0%	0.4–1.3%	1.3% vs. 1.7%	0.4% vs. 0.4%	2.1% vs. 2.5%	25% vs. 30%
COBIS	Registry	736	329	22	0.9–0.7%	0.6–1.3%	9.1–3.4%	NA	10–4.9%	NA
Yamawaki et al. [79]	Registry	132	124	36	0–0.1%	0–0%	12–5%	0–0%	15–7%	NA
Two stents										
Ge et al. [77]	Observa-tional	116	65	9	1.7–0%	10–14%	9.5–24.6%	2.6–3.1%	20–39%	24–38%
Grundeken et al. [76]	Registry	624	121	12	1.7–4.6%	5.0–4.6%	4.7–2.9%	0.3–0.9%	NA	NA

## Data Availability

No new data were created or analyzed in this study. Data sharing is not applicable to this article.

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
