# Peer review of "Current Challenges in Coronary Bifurcation Interventions"

_medicina, 2024, doi:10.3390/medicina60091439_

Round 1
Reviewer 1 Report
Comments and Suggestions for Authors
The article contains a lot of basic knowledge for publication in a scientific journal in 2024. In addition, the majority of references are very old.
Author Response
Dear Reviewer,
Thank you for your valuable feedback and for taking the time to review our manuscript. We appreciate your insights regarding the content and references.
We understand your concern about the foundational aspects of our article. We aimed to provide a comprehensive overview for readers who may be new to the topic, while also presenting original findings that contribute to the field. However, we recognize the importance of ensuring the manuscript meets the expectations for a scientific journal in 2024.
In response to your comment about the age of our references, we thoroughly reviewed the literature. We included more recent studies to enhance the manuscript’s relevance and depth. We believe this will help to ensure that our work is aligned with the current state of research and acknowledges the latest findings in the field. However, the references that are included and dated back to before the year 2000 are still important for our review as they concern the important yearly ages of bifurcation treatment.
Thank you once again for your constructive feedback, which undoubtedly improved the quality of our manuscript.
Reviewer 2 Report
Comments and Suggestions for AuthorsØ This review critically analyzes the unique characteristics and anatomical configuration of coronary bifurcations, focusing on the primary challenges encountered in the interventional management of these lesions. Even though the manuscript has been well written and organized, some changes need to be made. The text is not very readable.
The rationale and objectives of the study should be clearly stated at the end of the introduction.
Ø The manuscript contains numerous outdated references, with 30 citations dating back to before the year 2000, which should be replaced with more current sources.
Ø The use of the pronoun "we" in the fifth line of the abstract should be replaced with the phrase or It should be write as follows: “This review evaluates various methodologies designed to address these lesions, considering factors such as plaque distribution and bifurcation geometry. It also emphasizes the limitations associated with current techniques.”
Ø Extensive proofreading and rephrasing are essential.
In academic writing, paragraphs generally have between two hundred and three hundred words, although this can vary. It is necessary for paragraphs to fully develop so that all relevant information is covered within it. A new paragraph is thus introduced only when a new concept arises; therefore, most paragraphs should be divided into several parts.
Ø The conclusion section should be modified with additional key points of review.
Ø The plagiarism percentage is 30%, exceeding the recommended threshold of 20%.
Comments on the Quality of English Language
Author Response
- This review critically analyzes the unique characteristics and anatomical configuration of coronary bifurcations, focusing on the primary challenges encountered in the interventional management of these lesions. Even though the manuscript has been well written and organized, some changes need to be made. The text is not very readable.
The rationale and objectives of the study should be clearly stated at the end of the introduction.
Authors’ reply: Thank you for your thorough review and for your positive comments regarding our manuscript. We greatly appreciate your feedback, as it helps us improve the quality and clarity of our work. We acknowledge your observation concerning the readability of the text. We revisited the manuscript to identify areas where the language can be simplified and improved for clarity.
In response to your suggestion about articulating the rationale and objectives of the study more clearly at the end of the introduction, we completely agree. We included a concise statement that explicitly outlines the rationale for our review and the specific objectives we aim to achieve.
- The manuscript contains numerous outdated references, with 30 citations dating back to before the year 2000, which should be replaced with more current sources.
Authors’ reply: We appreciate your insights regarding the references of the manuscript. We thoroughly reviewed the literature, and we included more recent studies to enhance the manuscript’s relevance and depth. We believe this will help to ensure that our work is aligned with the current state of research and acknowledges the latest findings in the field. However, the references that are included and dated back to before the year 2000 are still important for our review as they concern the important yearly ages of bifurcation treatment.
- The use of the pronoun "we" in the fifth line of the abstract should be replaced with the phrase or it should be write as follows: “This review evaluates various methodologies designed to address these lesions, considering factors such as plaque distribution and bifurcation geometry. It also emphasizes the limitations associated with current techniques.”
Authors’ reply: Thank you for this useful suggestion. We have corrected the above sentence in the abstract with the one suggested by you.
- Extensive proofreading and rephrasing are essential.
Authors’ reply: Thank you for your constructive feedback. The manuscript has been proofread by a native speaker and we believe it has been significantly improved.
5.In academic writing, paragraphs generally have between two hundred and three hundred words, although this can vary. It is necessary for paragraphs to fully develop so that all relevant information is covered within it. A new paragraph is thus introduced only when a new concept arises; therefore, most paragraphs should be divided into several parts.
Authors’ reply: Thank you for your helpful comment. We completely agree with you that each new concept should be introduced in a subsequent paragraph. Therefore, we have extensively reworked our paragraphs and split them where necessary.
- The conclusion section should be modified with additional key points of review.
Authors’ reply: Thank you for your proposal. The conclusion paragraph has been modified as per your recommendation.
Ø The plagiarism percentage is 30%, exceeding the recommended threshold of 20%.
Authors’ reply: Thank you very much for this important feedback. We have thoroughly reviewed the manuscript and made extensive edits to improve the text and reduce the percentage of plagiarism.
Round 2
Reviewer 1 Report
Comments and Suggestions for Authors
Excellent work. All requested changes were addressed accordingly. It can be accepted for publication without further corrections.
Author Response
Excellent work. All requested changes were addressed accordingly. It can be accepted for publication without further corrections.
Authors: Thank you for your high remark!
Reviewer 2 Report
Comments and Suggestions for Authors
The authors tried to modify the manuscript. It has the potential to be published. However, they did not split paragraphs to increase readability. Some paragraphs are too long!
Comments on the Quality of English LanguageModerate proofreading is required.
Author Response
The authors tried to modify the manuscript. It has the potential to be published. However, they did not split paragraphs to increase readability. Some paragraphs are too long!
Author: Thank you for the time and effort spent on our manuscript. We have now modified the paragraphs and split them to increase readability.